# Triboelectric Nanogenerators for Efficient Low-Frequency Ocean Wave Energy Harvesting with Swinging Boat Configuration

**DOI:** 10.3390/mi14040748

**Published:** 2023-03-28

**Authors:** Jin Yan, Zhi Tang, Naerduo Mei, Dapeng Zhang, Yinghao Zhong, Yuxuan Sheng

**Affiliations:** 1Ship and Maritime College, Guangdong Ocean University, Zhanjiang 524088, China; 2Mechanical Engineering College, Guangdong Ocean University, Zhanjiang 524088, China; 3Shenzhen Research Institute of Guangdong Ocean University, Shenzhen 518120, China

**Keywords:** triboelectric nanogenerator, swinging boat-shaped structure, COMSOL, wave energy harvesting

## Abstract

To reach ocean resources, sea activities and marine equipment variety are increasing, requiring offshore energy supply. Marine wave energy, the marine renewable energy with the most potential, offers massive energy storage and great energy density. This research proposes a swinging boat-type triboelectric nanogenerator concept for low-frequency wave energy collection. Triboelectric electronanogenerators with electrodes and a nylon roller make up the swinging boat-type triboelectric nanogenerator (ST-TENG). COMSOL electrostatic simulations and power generation concepts of independent layer and vertical contact separation modes of operation explain the device functionality. By rolling the drum at the bottom of the integrated boat-like device, it is possible to capture wave energy and convert it into electrical energy. Based on it, the ST load, TENG charging, and device stability are evaluated. According to the findings, the maximum instantaneous power of the TENG in the contact separation and independent layer modes reaches 246 W and 112.5 μW at matched loads of 40 MΩ and 200 MΩ, respectively. Additionally, the ST-TENG can retain the usual functioning of the electronic watch for 45 s while charging a 33 µF capacitor to 3 V in 320 s. Long-term low-frequency wave energy collection is possible with the device. The ST-TENG develops novel methods for large-scale blue energy collection and maritime equipment power.

## 1. Introduction

With the exploitation of maritime resources, there is a need to integrate different electronic systems such as marine equipment and sensors through marine platforms or buoys for marine weather monitoring, signal transmission, and navigation, significantly increasing the need for energy. Marine monitoring sensors confront a significant obstacle of unsustainable power supply because traditional energy supply techniques often employ battery power; however, battery power has high cost and low range difficulties. The rise of nanogenerators provides a brand new idea to solve the problem of power supply for marine equipment, and its advantages such as its small size and good fatigue resistance prompt it to blossom in the marine business [1]. Nanogenerators are used not only in the marine field but also in the medical field. Their unique power generation method provides a valuable reference for the application of nanogenerators in the marine field [2,3,4,5,6]; nanogenerators use more non-metallic materials, and even the power generation part is all non-metallic materials, which reduces the weight and improves flexibility and fatigue resistance. Their power generation capacity is also very considerable. The energy used to drive the nanogenerators is also wide, among which wave energy is a potential contributor [7]. Wave energy is one of the marine renewable resources with the most potential because of its minimal environmental effect, high grade, and widespread distribution [8,9]. It is challenging to deliver direct power to offshore equipment using conventional wave energy producing systems due to drawbacks including their cumbersome construction and poor conversion efficiency. In order to keep vital offshore tools and sensors operational, it is essential to investigate alternative wave power generating methods.

Based on Maxwell’s displacement current theory, the triboelectric nanogenerator (TENG) is a revolutionary energy device that turns wasteful mechanical energy into electrical energy [10,11]. Due to the many benefits of TENG, such as its low cost, simple manufacturing, high power density, and the potential of self-provided energy monitoring, an increasing number of researchers [12,13,14,15,16,17,18,19,20,21] use TENG for energy harvesting and self-provided energy monitoring. When compared with the more traditional method of using electromagnetic induction for power production [22,23,24], TENG has been proven to be optimal for collecting energy at low frequencies (5 Hz). Vibrational energy, liquid droplets, water wave energy, and even human motion could be transformed into usable electrical power utilizing TENG [25,26,27,28,29,30,31], and low ocean wave energy is wave energy with longer wavelengths and lower frequencies in the ocean. This energy usually comes from natural forces such as wind, tides, and earth rotation in the ocean. This energy provides a clean and renewable energy source. Currently, however, it can be converted into usable electrical or other forms of energy through efficient low wave energy harvesting techniques. However, its share of the global energy supply remains low due to technical and economic constraints. Therefore, TENG is a great option for collecting the energy of low-frequency irregular waves.

Ocean wave energy is a kind of renewable energy, which will not be depleted like fossil fuels, its storage capacity is huge, its density is extremely high, and it can be used continuously. Collecting ocean wave energy does not produce pollutants, has no negative impact on the environment, and is a green energy. Using ocean wave energy can reduce the dependence on traditional energy sources and reduce non-renewable energy consumption and carbon emissions and has a greater role in mitigating the greenhouse effect. Ocean wave energy is a potential economic resource that can bring certain economic benefits to the local economy. Ocean wave energy can be collected in a variety of sea environments, such as oceans, bays, and straits, with wide applicability, and the collection of ocean wave energy is of great significance to sustainable development strategies and environmental protection strategies.

The Maxwell displacement current is the source of the basic TENG model [32]. In 1861, Maxwell projected the existence of electromagnetic waves and created the concept of displacement current to fulfill the equations of continuity of charge. This theory was validated by experiments in 1886, demonstrating its validity. The expressions of Maxwell’s system of equations are as follows [33]:(1)$$\nabla \cdot D=P_{f}$$
(2)∇⋅B=0
(3)$$\nabla \times E=-\frac{\partial B}{\partial t}$$
(4)$$\nabla \times H=J_{f}+\frac{\partial D}{\partial t}$$
where E,$\;P_{f}$, $J_{f}$, B, H, and D represent the electric field, free charge density, free current density, magnetic field, magnetization field, and displacement field, respectively.
(5)$$D=\varepsilon_{0}E+P$$
where $\varepsilon_{0}$ is the vacuum dielectric constant and P is the polarization field density.

For an isotropic medium $P=\left(\varepsilon -\varepsilon_{0}\right)E,\;D=\varepsilon E,$ and ε is the dielectric constant. The second term on the right-hand side of Equation (4) is defined as the Maxwell displacement current.
(6)$$\frac{\partial D}{\partial t}=\varepsilon_{0}\frac{\partial E}{\partial t}+\frac{\partial P_{S}}{\partial t}$$

The first term on the right side of Equation (6) is the induced current, which is generated by the changing electric field and is the theoretical basis for the existence of electromagnetic waves [34]. Nanogenerator theory and operation are grounded in the second term, which is the current induced by the polarization field produced by the electrostatic charge on the surface [33]. Depending on the mechanical energy to be captured, the TENG depends on various power generating concepts and hence constructs multiple power generation structures, as illustrated in Figure 1. The TENG might be operated in four different modes: vertical contact separation, horizontal sliding, single electrode mode, and independent layer mode [35].

Triboelectric nanogenerator (TENG)-based wave energy harvesting technology is attracting attention, and scientists are creating new TENG structures and testing out new TENG operating modes to maximize the system’s efficiency. There are now many different types of TENG that may be used for wave energy harvesting, such as liquid–solid mode TENG, totally enclosed TENG, spinning disc TENG, hybrid TENG, etc. Lin et al. [36] presented the first TENG operating in a liquid-solid phase. In 2013, the substrate’s electrodes were separated by a contact separation of PDMS and water; this created an electrical potential difference, which in turn drove electron movement in the load channel between the electrodes and produced electricity. In 2018, Li et al. [37] created a buoy-shaped liquid-solid TENG that can gather energy from various low-frequency vibrations. In 2020, Liu et al. [38] developed a planar-shaped power cable for a TENG network, which consists of a spring steel strip and a three-layer polymer film, and which maintains the stability of the TENG network. In 2018, Cheng et al. [39] transformed the hard-contact spherical TENG into the soft-contact spherical TENG (SS-TENG) by optimizing the material and structural de-sign, and the maximum output charge increased by a factor of ten compared to the standard TENG.Xu et al. [40] developed a high-power density triboelectric nanogenerator based on a tower structure in 2019 and experimentally proved that its power density could rise proportionately with the increase in the number of parallel units. For efficient energy harvesting from low amplitude waves, Zhang et al. [41] developed a Pelamis serpentine-based triboelectric nanogenerator (SS-TENG) with a lightweight, simple construction and cheap cost. Zhang et al. [42] developed integrated 12 sets of positive dodecahedral triboelectric nanogenerators with a multilayer wave-like structure (WS-TENGs). This device illustrates an effective method that is superior to standard water wave energy gathering methods. Xu et al. [43] created an integrated TENG array device in 2017 that uses air pressure to transmit and distribute the energy harvested from water waves using a new spring-suspended oscillator construction. In Ahmed et al. [44], a duck-shaped TENG was designed to efficiently capture energy from low-frequency and unpredictable water waves in 2016. Among the many, Jiang stands out. Triboelectric nanogenerator with wiggling structure and excellent energy conversion efficiency was invented in 2020 [45]. (SS-TENG). Triboelectric electric (TENG) and electromagnetic (EMG) hybrid systems based on improved internal topology were developed by Wang et al. [46] for efficient water wave energy harvesting in 2019. Wu et al. [47] developed a magnetic-sphere-based hybrid triboelectric electric (TENGs)-electromagnetic (EMGs) water wave energy harvesting system. This work proposes the design of a swinging ship-type triboelectric nanogenerator (ST-TENG) that operates in two modes: independent layer and contact separation. The ST-TENG has a nylon roller, an internal ship-type device, and an external ship-type device. Aluminum electrodes, foam adhesive, and PTFE film are used in the internal boat device, which was 3D printed. Additionally 3D printed, the exterior boat device has metal electrodes and PTFE film. Low-frequency disorganized wave energy is captured by the oscillation of the integrated boat form and the rolling of the nylon roller. Compared with many studies, this design has its own advantages. The electrode of this design is aluminum, which greatly reduces the cost and weight compared with the use of precious metal materials, and because of its light mass, the driving energy required is relatively small, so that it can generate relatively large power under weak wave energy. This design has no components that generate impact and need deformation to improve its service life. The simple structure of this designed product does not require high manufacturing precision, and the later maintenance is more convenient and adaptable to complex environments. The electrical theory simulation and calculation analysis of the ST-TENG in the two modes were performed using the COMSOL simulation software. The potential distributions under various motion positions were evaluated separately, establishing the theoretical foundation for the following physical fabrication, particular experimental scheme design, and comparative reference of experimental data, as well as recommendations for the triboelectric nanogenerator to harvest ocean wave energy and contribute to the collection of clean energy.

## 2. Structural Design and Power Generation Principle Analysis of the ST-TENG

### 2.1. ST-TENG Working Principle and Structure Analysis

The fundamental idea of a triboelectric nanogenerator (TENG) may be represented by two working modes, a vertical contact separation type and an independent layer mode, both of which were used in this research to design a swinging ship-type triboelectric nanogenerator (ST-TENG), as illustrated in Figure 2a,b, which consisted of three parts: a nylon roller (shown as 1 in Figure 2b), a built-in boat-shaped device (shown as 2 in Figure 2b), and an external boat-shaped device (shown as 3 in Figure 2b). The first class of triboelectric nanogenerators had a freestanding layer structure (F-TENG) that included a nylon roller with electrodes connected to the surface, polytetrafluoroethylene (PTFE), aluminum electrodes, foam adhesive, and an integrated boat-shaped device constructed of polylactic acid material. In order to generate an induced current in the external load connecting the two electrodes, the electric potential between the two electrodes on the inner bottom surface of the built-in boat device was periodically changed by rolling the nylon roller, which acted as a separate moving layer, from side to side. The outer bottom electrode of the device was separated from the Teflon electrode covered with another electrode by swinging the integrated boat-shaped device from side to side. As a result, a potential difference between the two electrodes was created, which resulted in the generation of an induced current. The polylactic acid utilized in the 3D-printed boat-shaped gadget was both biocompatible and biodegradable (PLA). The electrodes for aluminum electrodes could be constructed from various metal electropositive materials, such as copper film and silver film, etc., after the requirement was compared via experimentation. Foam adhesive has a certain degree of flexibility, and its function was to expand the contact area. Through the use of wave-driven drums and oscillating boat-shaped devices, the ST-TENG was able to efficiently capture low-frequency, irregular wave energy.

The F-TENG worked as shown in Figure 3. In response to external excitation (ocean waves) and rolling back and forth on the inner surface of the built-in boat-shaped device, the aluminum coating on the drum’s surface became positively charged. After numerous cycles of friction, the PTFE film became negatively charged, and the charge persisted on its surface for a long time since PTFE is electret. As shown in Figure 3a, when the drum rolled to the left above the electrodes, a negative charge was generated on the left electrode, and a positive charge was induced by the aluminum coating on the surface of the drum. When the drum was pounded from left to right (Figure 3b), the induced current flowed from right to left because the electrons in the external circuit flowed from left to right to compensate for the potential difference between the left and right electrodes. All negative charges were on the right electrode when the whole drum rolled over it (Figure 3c). When the drum rolled back to the left (Figure 3d), electron backflow occurred, causing a reverse current to be induced in the external circuit. Therefore, the drum could be rolled between the left and right electrodes thanks to the external stimulation, creating an alternating current (AC) in the external circuit.

Figure 4 depicts the operating principle shared by the second and third classes of triboelectric nanogenerators (CS-TENG). In response to external stimulation (such as ocean waves), the drum rolled back and forth on the inside surface of the boat-shaped device, causing the device to oscillate laterally as it did so. The aluminum electrode and the PTFE film acquired positive and negative charges, respectively, when the electrodes on the device’s exterior rubbed against the PTFE film over the course of multiple cycles. Figure 4a demonstrates how the aluminum electrode on the device’s outer surface came into contact with the negatively and positively charged surfaces of the PTFE when the built-in boat-shaped device swung to the left.

When the integrated boat device swung to the right (see Figure 4b), the aluminum electrode began to separate from the PTFE. At this point, electrons started moving from the electrode on the external boat device to the electrode on the interior boat device through an external circuit in order to balance the potential difference between the two electrodes. When the internal boat deice swung to the right with the maximum amplitude (Figure 4c), the maximum separation distance from the left electrode, all of the positive charge was transferred to the electrode inside the external boat device. When the internal boat device swung back to the left, a reverse current was created due to electron backflow in the external circuit (Figure 4d). Thus, under external excitation, the built-in boat-shaped device could oscillate reciprocally so that the two electrodes generated an alternating current in the external circuit.

### 2.2. Two Models of ST-TENG Theoretical Basis

Figure 5 depicts the theoretical model of the first group functionally independent layer triboelectric nanogenerator (F-TENG). Different materials may be employed as independent layers, allowing the sliding independent layer model to be classified as either dielectric or metallic; the latter is the one utilized here, wherein metals 1 and 2 were employed as electrodes, with a distance g between them, and wherein a separate layer of metal of size l was positioned above this plane, with a distance h between the bottom surface of the separate layer and the metal. The potential at the dielectric’s base could not be considered a node since it varied. Here it was assumed that a region of the dielectric material’s surface, denoted by dk, carried a frictional charge with a charge density of -*σ* and that the combined charge of the two metals was σwdk; this was studied in terms of potential superposition. In the case of a short circuit, the charges of the two electrodes (*dQ*_1_ and *dQ*_2_) can be represented by the following equation, and the capacitance between the small interface and the metal *i* can be represented by *Ci*(*k*).
(7)$$dQ_{1}=\frac{\sigma wdk}{1+\frac{C_{2}(k)}{C_{1}(k)}}$$
(8)$$dQ_{2}=\frac{\sigma wdk}{1+\frac{C_{1}(k)}{C_{2}\left(k\right)}}$$

The total charge of the two electrodes is described by the following equation based on the principle of superposition of potentials because the total charge on the surface of the dielectric material was superimposed by all electrostatic regions:(9)$$Q_{1}=\sigma w\int_{0}^{l}\frac{dk}{1+\frac{C_{2}(k)}{C_{1}(k)}}$$
(10)$$Q_{2}=\sigma w\int_{0}^{l}\frac{dk}{1+\frac{C_{1}(k)}{C_{2}(k)}}$$

Therefore, *Q_SC_* finally can be expressed by the following equations:(11)$$Q_{SC,final}=\int_{0}^{l}\frac{\sigma wdk}{1+{\left(\frac{C_{2}(k)}{C_{1}(k)}\right)}_{x=g+l}}-\int_{0}^{l}\frac{\sigma wdk}{1+{\left(\frac{C_{2}(k)}{C_{1}(k)}\right)}_{x=0}}$$

When *x* = 0, the distance between surface of dielectric material and metal 1 will be much greater than the distance between metal 2, so the ratio *C*_2_(*k*)/*C*_1_(*k*) approaches 0 when *k* is any value. From Equations (9) and (10), we can see that *Q*1 approaches *σwl* and *Q*_2_ approaches 0. In contrast, when *x* = *g* + l, the *C*_2_(*k*)/*C*_1_(*k*) ratio tends to infinity regardless of the value of *k*, so *Q*_1_ approximates to 0, and *Q*_2_ approximates to *σwl*. Theoretically, the *Q_SC_* can reach *σwl*, and therefore, the transfer charge efficiency (*ŋ_CT_*) can reach 100%. The variations in the capacitance ratio and x regulate the passage of electrons between two electrodes. Here the capacitance between the electrodes and *Q_SC_*(*x*) did not have a suitable analytical formula due to the boundary effect, and subsequent simulations can analyze the exact principle.

Figure 6 depicts the theoretical model of the vertical contact separation triboelectric nanogenerators (CS-TENG). The theoretical model of the contact separation TENG can be divided into two categories based on the various options for friction materials. These categories are the contact friction type between two dielectric materials and the contact friction type between the dielectric material and the conductive material. This research used the theoretical model of friction, in which the conductive material with associated electrodes and the dielectric material acting as the friction layer were stacked face to face. The dielectric material thickness is *d*1, and its relative dielectric constant is *εr*. The distance *x* between the dielectric and conductive materials can change depending on the excitation received.

When the two materials come into contact owing to the excitation, the inner surface will take on a charge with a charge density of due to the different positions and electronegativity of the two in the frictional electric sequence, although the two have opposite signs. Furthermore, it could be assumed that the surfaces of the conducting and dielectric materials have a uniform distribution of frictional charge and that the charge density does not drop down fast. The dielectric and conducting materials are separated from one another by a distance x after the stimulation fades. The charge transfer may be characterized as Q, and the induced potential difference (V) between the two electrodes is what creates the flow of electrons in the external circuit between the two electrodes. As a result, the instantaneous charges on the two electrodes are *Q* and −*Q*, respectively.

The area of the two electrodes is considered to be infinite under typical experimental settings due to the fact that the electrode size (*S*) exceeds the separation distance (*d* + *x*) between conducting and dielectric materials. The real-time power generation performance of TENG is described by the *V*-*Q*-*X* relationship, where *V* and *Q* are the voltage and charge transfer between electrodes, respectively, and x is the distance between two friction materials. Figure 6a depicts a physical model from which these three correlations can be derived using standard electrodynamic theory. Since the area of an electrode can be regarded limitless, its electrons can be assumed to be uniformly dispersed along its inner surface. In the region between the dielectric and conducting materials and within the dielectric itself, electric field lines only exist perpendicular to the plane. According to Gauss’ theorem, the electric field strength corresponding to each region can be expressed as follows:

Within the dielectric material 1:(12)$$E_{1}=-\frac{Q}{S\varepsilon_{0}\varepsilon_{r}}$$

Inside the air gap:(13)$$E_{air}=-\frac{-\frac{Q}{S}+\sigma (t)}{\varepsilon_{0}}$$

The potential difference between the two electrodes can be expressed as follows:(14)$$V=E_{1}d_{1}+E_{air}x$$

The V-Q-x relationship for this theoretical model of the triboelectric nanogenerator can be obtained as follows:(15)$$V=E_{1}d_{1}+E_{air}x=-\frac{Q}{S\varepsilon_{0}}\left(\frac{d_{1}}{\varepsilon_{r}}+x(t)\right)+\frac{\sigma x(t)}{\varepsilon_{0}}$$

The ratio of the thickness of the dielectric material to its relative permittivity is defined as the effective thickness constant *d*_0_, at which point the $V_{oc}$, *Q_SC_*, and *C* of the triboelectric nanogenerator can be derived as follows:(16)$$V_{OC}=\frac{\sigma x(t)}{\varepsilon_{0}}$$
(17)$$Q_{SC}=\frac{S\sigma x(t)}{d_{0}+x(t)}$$
(18)$$C=\frac{\varepsilon_{0}S}{d_{0}+x(t)}$$

In addition to the method described above, the V-Q-x relationship can also be determined by a particular electrostatic induction mechanism. In this method, each node is defined as an equipotential surface or body, whose area can be viewed as infinite in comparison with the thickness of the dielectric material. As a result, the entire surface of the dielectric material is treated as node 2. The inner surfaces of electrodes 1 and 2 are regarded as nodes 1 and 3, as indicated in Figure 6b, because the metal electrode surfaces are all equally potential. Two nodes can be thought of as having an equivalent capacitance because electric field lines will exist between them. However, as the dielectric material is assumed to have an unlimited surface area, node 2 will act as a shield for the electric field lines between nodes 1 and 3, resulting in a system with two capacitances. In the short-circuit case, node 1 and node 3 will become an equipotential surface; at this time, the charge of node 2 is -σS, and the total charge of the two nodes is σS. Based on Kirchhoff’s law and the conservation of charge, the charge on node 3 can be found as
(19)$$Q_{node3}=\frac{\sigma S}{1+\frac{C_{1}(x)}{C_{2}\left(x\right)}}$$

When adding from 0, the short-circuit transfer charge (*Q_SC_*) from node 3 to node 1 can be expressed as follows:(20)$$Q_{SC}=\frac{\sigma S}{1+\frac{C_{1}(x)}{C_{2}\left(x\right)}}-\frac{\sigma S}{1+\frac{C_{1}(x=0)}{C_{2}\left(x=0\right)}}$$

According to the parallel plate capacitance model, Equation (20) can be expressed in the same way as Equation (17). As seen in the preceding derivation, adjusting the location of the charged plane alters the capacitance ratio *C*_1_*/C*_2_, which in turn alters the charge transfer between the two electrodes in the short-circuit state. When x is large enough, we define the charge transfer efficiency (i.e., the ratio of the transferred charge to the total frictional charge)$\;{ŋ}_{CT}$ as
(21)$${ŋ}_{{}_{CT}}=\frac{Q_{SC}{}_{,\mathrm{fin}al}}{\sigma S}=\frac{1}{1+\frac{C_{1}(x=x_{\max})}{C_{2}\left(x=x_{\max}\right)}}-\frac{1}{1+\frac{C_{1}(x=0)}{C_{2}\left(x=0\right)}}$$

For the triboelectric nanogenerator with the operating mode of vertical contact separation, *C*_1_ (*ε*0*S*/*d*0) approached infinity when the friction materials were in contact with each other (*x* = 0), while C_2_ was a finite value, at which point Qnode3 = 0. When the friction materials were far enough apart (*x* > 10*d*), C_1_ (*ε*0*S*/*d*0) approached 0, while C_2_ remained *ε*0*S*/*d*0, at which point Qnode3 = *σS*. Therefore, the maximum charge transfer efficiency in this mode of operation can theoretically reach 100%.

From the above derivation, some basic characteristics of the contact separation type of triboelectric nanogenerator can be seen: (1) The open-circuit voltage was linearly dependent on the friction material’s separation distance x, while the internal capacitance was proportional to the opposite quantity. (2) With growing distance between components, the short-circuit transfer charge $Q_{SC}$ saturated (*σS*). When the separation distance increased from 0 to 10 d0, the short-circuit transfer charge $Q_{SC}$ sharply reached 0.9*σS*. (3) In order to enhance the efficiency of charge transfer, the separation distance between the dielectric and the friction material should be minimized to zero. When the minimum separation distance varied around 10 d0, *C*_1_(*x*) (*ε*0*S*/*x*) was always smaller than *C*_2_(*x*) (*ε*0*S*/*d*0). The ratio *C*_1_*/C*_2_ approached zero and essentially stopped changing with more electrode separation, indicating that very little charge transfer was occurring between the electrodes. Furthermore, the theory was based on the capacitance indicated by the flat plate infinity theory, and if the separation distance x was close to the material size, the overall device performance was affected by boundary effects. Then, the open-circuit voltage was no longer linearly related to the separation distance.

### 2.3. ST-TENG Two-Mode Electrostatics Simulation Analysis

The study designed the swinging ship-type triboelectric nanogenerator, partitioned the mesh, and simulated the potential distribution of the friction material under different situations using the COMSOL simulation software, which specializes in multi-physics field coupling. The COMSOL simulation software was used to create a two-dimensional model of the F-TENG shape, as shown in Figure 7. Next, various materials were added to the material library for different geometries; for example, PTFE was used as the dielectric material, with a thickness of 1 mm and a Poisson’s ratio of 0.3. The independent drum’s surface material and the two electrode properties were uniformly set to aluminum, the width of both electrodes was set to 1 mm, the radius of the independent drum was set to 3.5 mm, the surrounding material was set to air dielectric, and the relative dielectric constant was 1. Furthermore, it should be noted that a gap of 2 mm must be left between electrode 1 and electrode 2 to ensure the flow of electrons in the external circuit between the two electrodes. In addition, a steady-state computation was used in the simulation procedure, assuming that the aluminum on the surface of the separate drum remained in complete contact with PTFE the entire time. In order to obtain the trend of potential change, the PTFE surface charge density was set to −0.01 μC/m^2^, and the total charge density on the aluminum surface was set to 0.01 μC/m^2^.

Figure 8 shows the steps used to mesh the geometry after the geometric model and material settings were completed. The mesh was initially predefined and set to “finer,” and the free triangular mesh was chosen for all materials. Because air had less influence on the induction potential distribution of this geometric model, the air domain mesh was set to “finer,” with a maximum cell size of 1 mm and a minimum cell size of 5 × 10^−3^ mm. The electrodes, metals, and dielectrics that were in contact with one another were all “ultra-fine,” with a cell size of 0.04 mm at their largest and 3 × 10^−4^ mm at their smallest. It can be seen in the figure that the mesh size was larger in air and decreased as one approached the electrodes, the specific metals, and the dielectric materials that formed the contact components.

Figure 9 depicts the result of solving Maxwell’s control equations using finite element simulation to determine the potential distribution over the two induction electrodes. As the roller passed above the left induction electrode, the charge on that electrode and the free metal was almost equivalent to that on the PTFE film (Figure 9a), resulting in a potential difference between the two electrodes. The potential difference between the two electrodes was at its maximum at this point, as can be seen in the simulation cloud diagram. By the time the rolling action reached its midpoint (Figure 9b), the charges of the electrodes on either side were almost equal, and the potential difference was minimal. When the roller passed over the right electrode, the potential difference reached its maximum value again but this time in the opposite direction (Figure 9c). When the roll returned to its center position (Figure 9d), the potential difference was at its lowest. When a potential difference was maintained, electrons were compelled to pass through the external circuit, generating a current (the details of electron transfer are shown in Figure 10). Based on the numbers, the potential difference can be as high as 155 V. As a result, the voltage needs of most current marine equipment and marine monitoring sensors are fulfilled by the simulation results.

In the same way, the geometric model of the second group of contact separation triboelectric nanogenerators was initially established in the COMSOL simulation software. In this case, the model was simplified to facilitate mesh division and reduce simulation computation because this group of triboelectric nanogenerators utilized a contact separation mode of operation. The material was assumed to be in a flat plate shape without considering its bending state. Figure 11 depicts the initial addition of the various components, with PTFE (the dielectric) set at 1 mm in thickness, aluminum (the electrodes) set at 1 mm in width, and air (the surrounding material) set at a relative dielectric constant of 1. Meanwhile, the PTFE surface charge density was set to −0.01 μC/m^2^, and the aluminum surface total charge density was set to 0.01 μC/m^2^.

Second, after the geometric model was built and the material was set, the geometric mesh was divided as shown in Figure 8, which shows four states: Al electrode fully contacted with PTFE (Figure 12a), separation startup (Figure 12b), maximum separation distance (Figure 12c), and when the approach is backward (Figure 12d). The free triangular mesh was chosen, and the mesh predefinition was set to “finer”. The maximum and minimum cell sizes for the air domain mesh were 1 mm and 5 × 10^−3^ mm, respectively. The air domain mesh was set to “finer”. The maximum and minimum cell sizes for the mesh of the two electrodes and the dielectric material were set to “superfine,” 0.04 mm, and 3 × 10^−4^ mm, respectively. The maximum and minimum cell sizes for the mesh of the two electrodes and the dielectric material were set to “ultra-fine”, 0.04 mm, and 3 × 10^−4^ mm, respectively.

Finally, the potential distribution of the two electrodes at different spacings could be calculated by Maxwell’s control equations using finite element simulation, as shown in Figure 13. When in the initial state, charge transfer occurred between the electrodes because the aluminum electrode was in full contact with the PTFE (Figure 13a). When the two began to separate (Figure 13b), an electrical field was generated between the electrodes and the air gap. This produced an electric potential difference between the upper and lower electrodes, and the potential cloud diagram showed that this potential difference developed steadily as the distance increased. This was in good agreement with Equations (2)–(10), which stated that the potential difference varied linearly with separation distance.

Figure 13c depicts the highest potential difference that can exist when the separation distance is at its greatest. The potential difference reduced when the separation distance was decreased from its maximum (Figure 13d). A uniform electric field was maintained if the separation distance was not too great in the simulation process; otherwise, the edge effect would have a significant effect, approximating the electric field line as the electric field line between two point charges. The details of the contact separation TENG electronic movement are shown in Figure 14.

## 3. ST-TENG Experimental Platform Construction and Device Fabrication

The experiments detailed in this paper involve fabricating the ST-TENG device, simulating waves, and data acquisition from the electrical signals generated as a result. The experimental research system to investigate the features of the ST-TENG power production is introduced, as well as the experimental plan and the various experimental equipment used in the experiments.

### 3.1. Experimental Equipment

We used a Keithley 6514 electrostatic meter. The input impedance up to a 200 TΩ high sampling rate could achieve dynamic current voltage and charge real-time acquisition. Combining the high-speed voltage acquisition card and system software meant that the highest speed of the collected signal could reach 50,000 points per second. The voltage and current measurement ranges of 10 μV to 200 V and 1 fA to 20 mA were ideal for the high-output-voltage triboelectric nanogenerator and low-output electrical current characteristics.

The data acquisition equipment that we used was as follows: The NI-9215 voltage acquisition card had four-channel simultaneous acquisition, a 100 KS/S acquisition rate per channel, 16-bit resolution, integrated signal conditioning, and a USB connection to the computer without external power. For the data acquisition process to work, the electrostatic meters must convert the analogue signal into a digital signal, which was then sent to the voltage acquisition card. The measurement data shown on the computer were connected to the voltage acquisition card.

The linear motor used in the experiment was a LinMot-E1200-RS type, and it simulated the motion of a wave with a maximum speed, thrust, and stroke of 3.2 m/s, 63 n, and 360 mm, respectively. To achieve high-speed and high-frequency linear motion, the supporting programmable control software could be employed to regulate the acceleration, speed, and displacement. The linear motor adequately agreed with the requirements of this experiment and satisfied the wave’s low-frequency and low-amplitude characteristics.

In this case, a professional-grade 3D printer, the Ultimaker-3, was employed. This model has a dual-nozzle design, ultra-long uptime, water-soluble support, and rapid nozzle changeover, all of which contributed to a cohesive, interconnected, and adhering 3D printing environment. The printing medium was PLA (Polylactic Acid), which has a wire diameter tolerance of 0.03 mm, melting point of 160–165 °C, and tensile strength of 250 kg/cm^2^. The internal and external ship structure was modeled by SOLIDWORKS, then imported into CURA software for slicing, and finally printed using the 3D printer.

### 3.2. ST-TENG Device Fabrication

Table 1 shows the experimental materials used in this experiment.

Initially, using the SOLIDWORKS modelling program, we created the blueprints for the inner and outer boat gadget, with the dimensions (205 × 77 × 75) mm in length, width, and height. Outside the boat, the device dimensions were (235 × 83 × 85) mm in length, width, and height. When the inner and outer boat devices were completed, the model was imported into CURA software for the slicing process. Next, a 2 mm thick boat device, including an inner and outer layer, was printed. For the purpose of this experiment, we prepared two types of rollers: a solid nylon rod of 60 mm diameter and 70 mm length and a PLA roller made via 3D printing. Despite having the same dimensions, the PLA roller was much lighter in weight than the nylon rod. By comparison, it was possible to evaluate the impact of various roller weights on the ST-power TENG’s generating properties. First, measure the same size of foam adhesive with the internal size of the built-in boat device, then apply two pieces of aluminum foil 10 mm apart to the foam adhesive by the bonding method to form two electrode layers, then use the same method to put a layer of polytetrafluoroethylene (PTFE) film on the surface of the electrode layer, and finally, put the whole device inside of the built-in boat and add the roller to form the first group of independent layer mode friction nanogenerators.

The second and third groups of triboelectric nanogenerators were put together using a similar method. PTFE film was bonded to the aluminum electrode surface by the bonding method after two pieces of aluminum foil were cut using the cutting method, forming a whole that was then inserted on both sides inside the external boat-shaped device. Second, the external bottom surface of the internal boat-shaped device had a specific size of electrode aluminum pasted on it. The assembled built-in boat device was placed within the exterior boat device to produce a complete ST-TENG, as illustrated in Figure 2a. The top of the exterior boat device was printed using a 3D printer in order to minimize the impact of the air and waves outside. A hot-melt glue gun was used to form a sealed state after being assembled and then enclosed the external boat device. The built-in boat device could swing from side to side, and the roller inside of it could roll back and forth owing to the wave drive. For optimum utilization of wave energy, it was essential for all three sets of triboelectric nanogenerators to reach the operating condition simultaneously.

### 3.3. Experimental Protocol Design

The experimental setup and the flow system is shown in Figure 15.

The optical plate and linear motor, which were fastened to one another, were connected via the initial connection. After mounting the ST-TENG on the push plate of a linear motor, a second connection was used to link the ST-TENG to the motor, which was subsequently controlled by the computer. As the linear motor software LinMot-Talk only allowed for the setting of acceleration, speed, and displacement, it was required to compute the displacement and acceleration at various frequencies and amplitudes by deriving the linear motion formula. Additionally, the data were gathered by connecting the positive and negative electrodes to the positive and negative contacts on the Keithley 6514 electrostatic meter. The Keithley 6514 converted the received digital signals, such as voltage, current, and charge, into analogue signals and transmitted to the NI voltage acquisition card, which then transmitted the received signals to the computer connected by the USB port. The data were gathered using the LabVIEW software, and the sampling frequency was set to 500 in order to save the data. The data were then processed and analyzed to provide the experimental results.

First, a simulation study using the COMSOL software was used to confirm the viability of the ST-TENG theory based on the well-established ST-TENG theoretical model. The experiments were conducted to examine the impact of various influencing variables on the power production characteristics of the ST-TENG. It was critical to account for both the internal and exterior elements, with the external ones such as the wave amplitude and frequency being of primary importance. The dielectric material, electrode size, and drum settings constituted the majority of the internal factors. The drum parameters could be subdivided into the drum material and drum weight, and the effect factors on the dielectric material could be divided up into the dielectric material type and dielectric material thickness. The control variable approach was used to investigate the effects of each element on the power production characteristics of the ST-TENG. After the comparison study, load characteristics, electrically brilliant LED lights, capacitor charging, and stability tests were undertaken before the ST-TENG was essentially enhanced. The feasibility of the ST-TENG was tested experimentally.

## 4. ST-TENG Important Characteristics Study

### 4.1. ST-TENG Load Characteristics Study

To learn more about how the ST-TENG works, it is important to examine the relationship between its external load and output power. The ST-TENG was connected with 100 KΩ to 2 GΩ resistors in order to regulate the external operating state and set the linear motor settings to 2.5 Hz frequency and 60 mm amplitude, respectively. As seen in Figure 16 and Figure 17, the output power could be computed using the formula P = I^2^R. Figure 17 demonstrates that a short-circuit condition could be seen when the external load was less than 10 MΩ since the external load was significantly less than the internal resistance of TENG at this time, the current was a short-circuit current, and the power grew slowly.

When the external load was between 10 MΩ and 1 GΩ, the external resistance was very comparable to the internal resistance of the CS-TENG. At this point, an increase in resistance would slow down the pace at which charges were transferred, causing the output current to decrease rapidly and the output power to increase quickly, reaching its maximum instantaneous power of 246 μW at 40 MΩ. The charge transfer would be fully inhibited as the external load rose, resulting in a very low current. Moreover, when the highest instantaneous power of the F-TENG was 112.5 μW, as shown in Figure 17, the corresponding internal resistance was 200 MΩ. It was evident that the output power of the ST-TENG was higher than other reports in the literature [48,49] and proved that the ST-TENG at low frequencies was capable of driving small smart electronic devices [50,51].

### 4.2. Study of the ST-TENG Charging Characteristics

In order to verify the power generation capability of the ST-TENG and its ability to be stored and utilized, this paper describes an experimental verification analysis at a frequency of 1 Hz and an amplitude of 50 mm, as shown in Figure 18, to first verify the electrical energy output of the F-TENG by connecting the electrode ends of the F-TENG to the rectifier through wires and charging capacitors with different capacitances after rectification. The experimental results showed that the F TENG could charge 3.3 µF and 10 µF capacitors to 9.26 V and 2.93 V, respectively, within 180 s (shown in Figure 19). Second, to verify the output of the CS-TENG, two groups of CS-TENGs were connected in parallel and then connected to the rectifier for rectification, and then different capacitors were charged (Figure 20); the experimental results showed that the CS-TENG could charge 1 µF and 3.3 µF capacitors to 11.16 V and 6.75 V, respectively, in 90 s (Figure 21). Finally, after connecting the CS-TENG and F-TENG in series and then conducting experiments on LED lighting and charging the capacitor after rectification before powering the electronic watch (shown in Figure 22), the experimental results showed that the device could charge the 33 µF capacitor to 3 V in 320 s (shown in Figure 23), which could light at least 100 LEDs. When the capacitor was connected to both ends of the electronic watch, it could be used to operate the electronic watch for 45 s. The experiment showed that the device had a certain output capacity and could be stored and used, but because the current was AC current and the current output was small, the LEDs could not be lit at the same time and could not reach the maximum brightness, so one could observe the lighted situation by the naked eye but could not photograph all the lit LEDs at the same time.

### 4.3. ST-TENG Stability Study

The durability of the ST-TENG must be investigated since its long-term reliability is crucial for the device to reliably gather wave energy. The amplitude and frequency were set to 1 Hz and 50 mm by configuring the linear motor parameters. The experiment was divided into three weeks and conducted once a week for 3000 cycles, simultaneously. Figure 24 shows that after three separate experiments, the ST-output TENG’s capacity was almost unchanged, indicating the device was stable and sustained low-frequency wave energy collection.

## 5. Conclusions

A swinging boat-type triboelectric nanogenerator containing an inner and outer boat-type device was designed, integrating two operating modes, independent layer and contact separation, and three sets of power generation units, which fully improved the structural utilization. Through structural analysis and theoretical derivation, the power generation principle of the ST-TENG was explored. The electrostatic theoretical simulation and calculation analysis were carried out by the COMSOL simulation software to obtain the potential distribution at different motion positions, which provided a theoretical basis for the design of specific experimental schemes and the comparison reference of the experimental results. A 3D printer printed the designed inner and outer boat-shaped devices. The commercial films used were combined by cutting and bonding methods to form a triboelectric nanogenerator that could perform two modes of operation simultaneously. By manipulating the acceleration, displacement, and velocity parameters, the experimental system was utilized to examine the effects of internal structural characteristics and external operating circumstances on the ST-TENG. Three basic ST-TENG features were investigated. For the load characteristics, the output current decreased with increasing external resistance, and the output power increased before reaching the best matching resistance and then decreased gradually. For the charging characteristics, the ST-TENG could charge a 33 µF capacitor from 0 to 3 V in 320 s and then power the electronic watch and maintain its normal operation for 45 s. In addition, the ST-TENG could light up at least 100 LEDs. The experimental results showed that the output capability of the ST-TENG remained unchanged under three experiments for three weeks, indicating excellent device durability to effectively collect low-frequency irregular wave energy for a long time.

## Figures and Tables

**Figure 1 micromachines-14-00748-f001:**
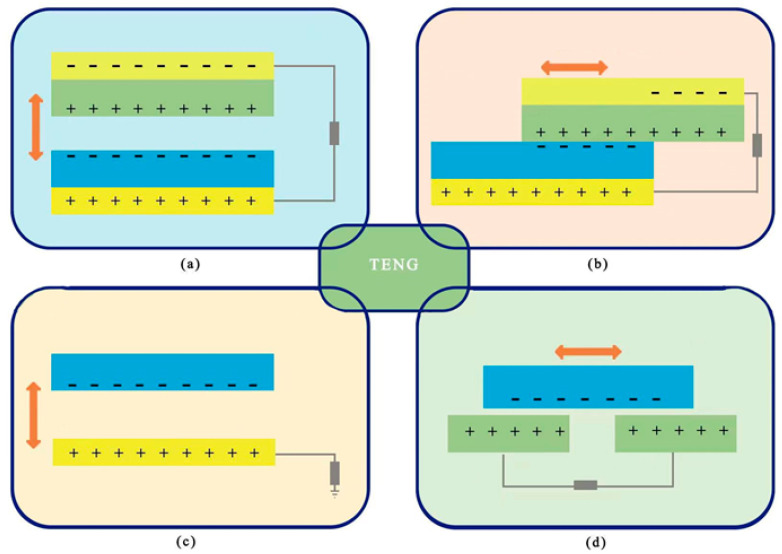
The four working modes of the TENG: (**a**) vertical contact separation type, (**b**) horizontal sliding type, (**c**) single electrode mode, and (**d**) independent layer mode.

**Figure 2 micromachines-14-00748-f002:**
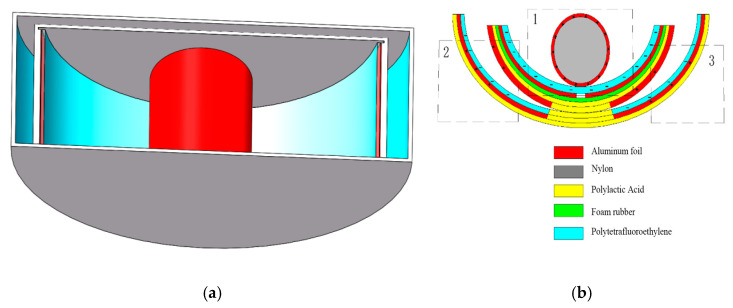
Swinging boat-shaped triboelectric nanogenerator structure. (**a**) The 3D model; (**b**) The structure section view.

**Figure 3 micromachines-14-00748-f003:**
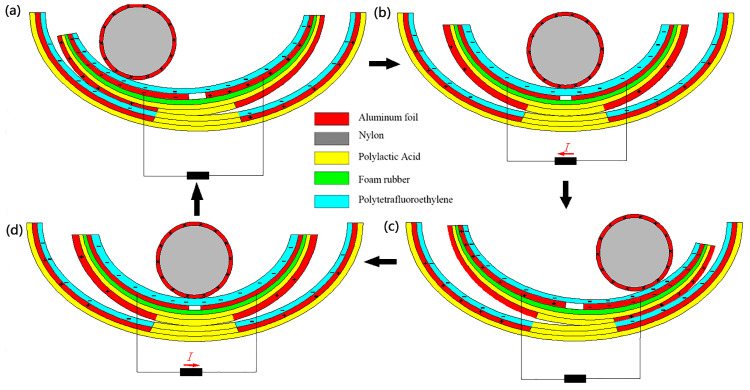
F-TENG working principle, (**a**) is when the independent drum rolls to the left above the electrode, (**b**) is when the drum rolls from left to right, (**c**) is when the whole drum rolls to the right above the electrode, (**d**) is when the drum rolls to the left again.

**Figure 4 micromachines-14-00748-f004:**
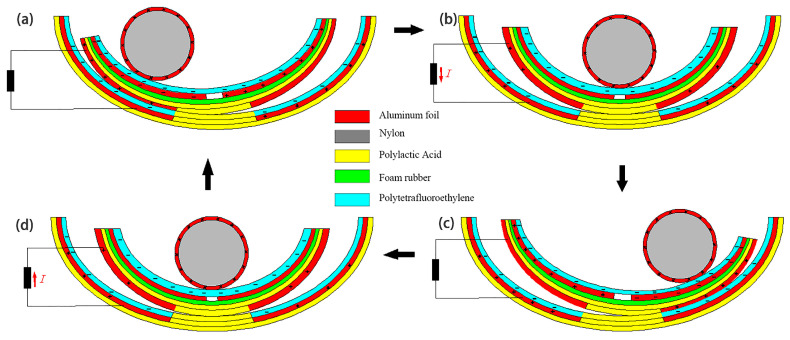
CS-TENG working principle. (**a**) is when the independent drum rolls to the left above the electrode, (**b**) is when the drum rolls from left to right, (**c**) is when the whole drum rolls to the right above the electrode, (**d**) is when the drum rolls to the left again.

**Figure 5 micromachines-14-00748-f005:**
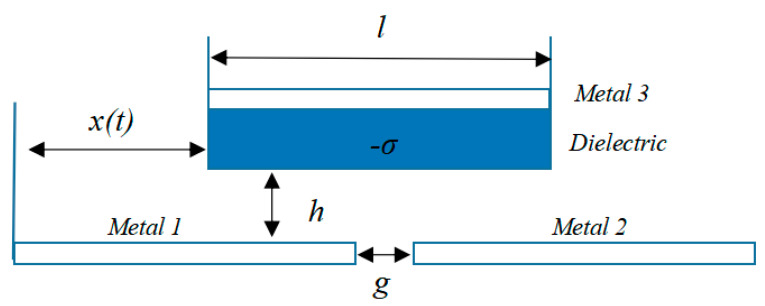
Theoretical model of the independent layer mode triboelectric nanogenerators.

**Figure 6 micromachines-14-00748-f006:**
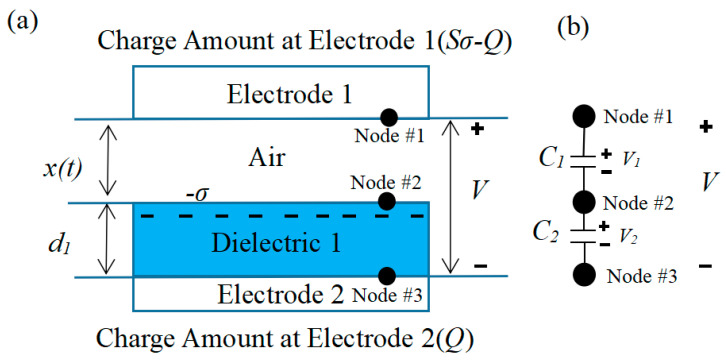
Theoretical model of contact-separated triboelectric nanogenerators. (**a**) the theoretical model, (**b**) the equivalent circuit diagram.

**Figure 7 micromachines-14-00748-f007:**
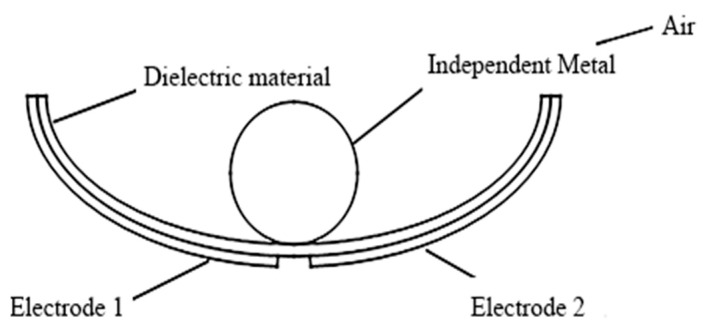
Freestanding TENG geometry model.

**Figure 8 micromachines-14-00748-f008:**
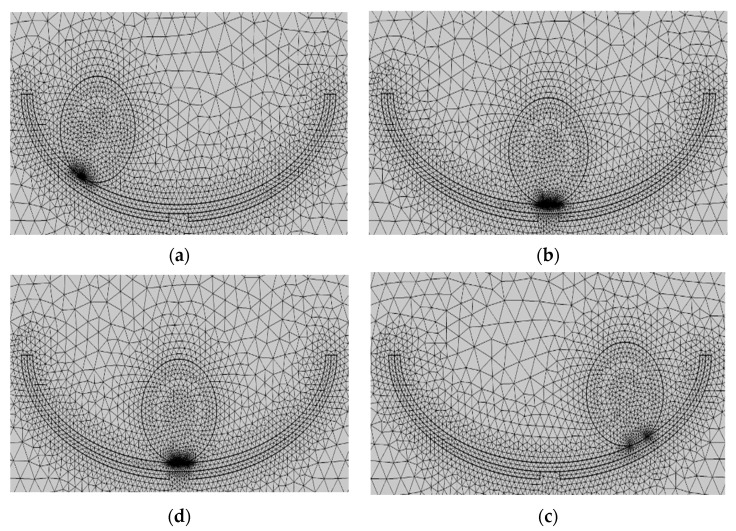
Freestanding TENG geometry meshing. (**a**) is the mesh division when the drum rolls over the left electrode, (**b**) is the mesh division when the roll moves to the middle position, (**c**) is the mesh division when the drum moves over the right electrode, and (**d**) is the mesh division when the roll roll rolls back to the middle position.

**Figure 9 micromachines-14-00748-f009:**
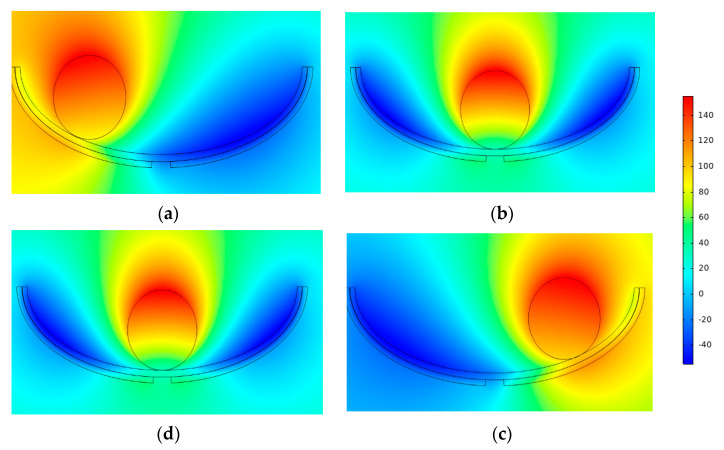
Stand-alone TENG simulation potential distribution. (**a**) is the potential difference when the drum rolls above the left electrode, (**b**) is the potential difference when the roll moves to the middle position, (**c**) is the potential difference when the drum moves above the right electrode, and (**d**) is the potential difference when the roll roll rolls back to the middle position.

**Figure 10 micromachines-14-00748-f010:**
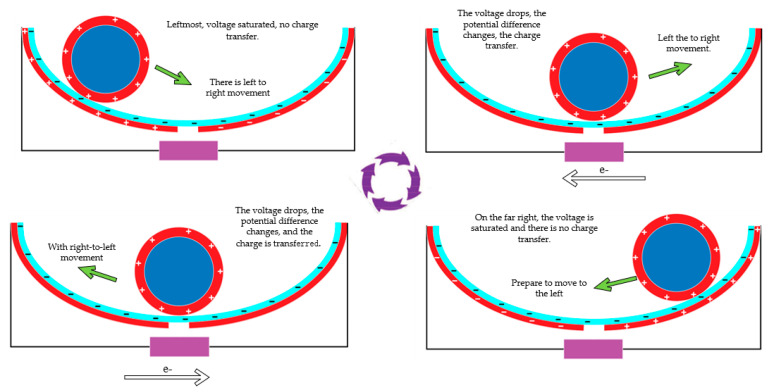
Stand-alone TENG charge movement diagram.

**Figure 11 micromachines-14-00748-f011:**
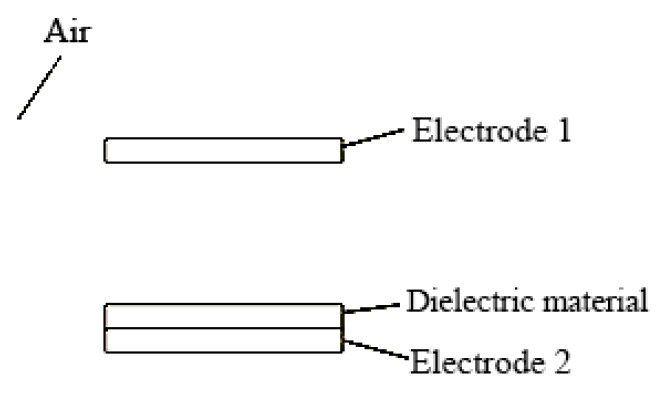
Contact-separated TENG geometry model.

**Figure 12 micromachines-14-00748-f012:**
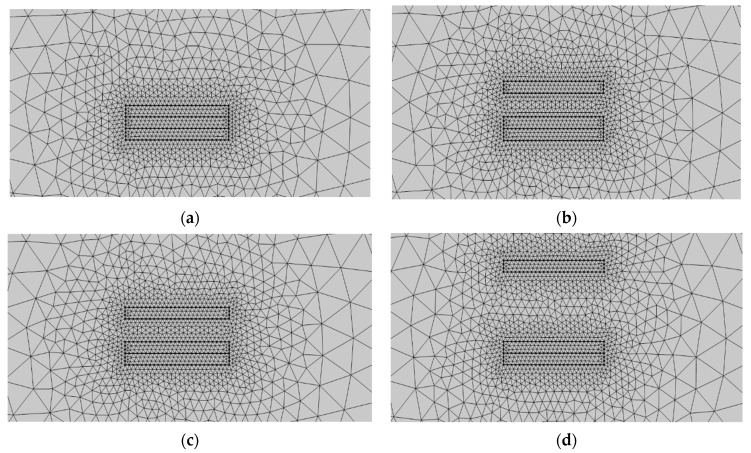
Contact-separated TENG geometry meshing. (**a**) is the mesh division when the Al electrode is in full contact with PTFE, (**b**) is the mesh division when the two start to separate, (**c**) is the mesh division at the maximum separation distance, and (**d**) is the mesh division when approaching backward.

**Figure 13 micromachines-14-00748-f013:**
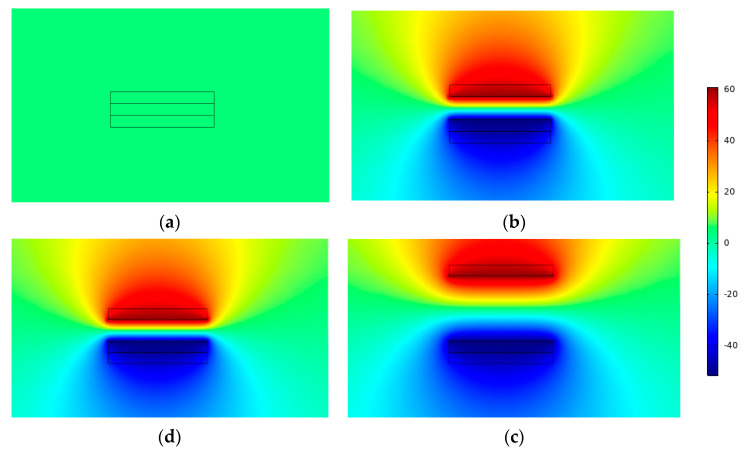
Contact-separated-type TENG simulation potential distribution chart. (**a**) is the potential difference when the Al electrode and PTFE are in full contact, (**b**) is the potential difference when they start to separate, (**c**) is the potential difference at the maximum separation distance, and (**d**) is the potential difference when approaching backwards.

**Figure 14 micromachines-14-00748-f014:**
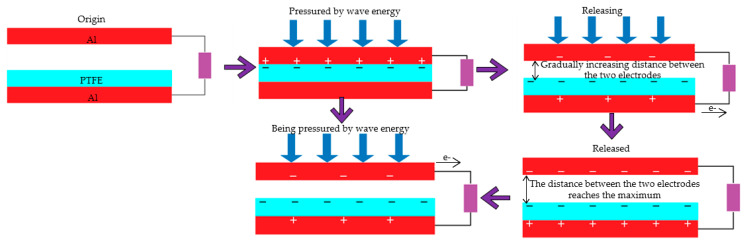
Contact separation TENG electronic movement situation.

**Figure 15 micromachines-14-00748-f015:**
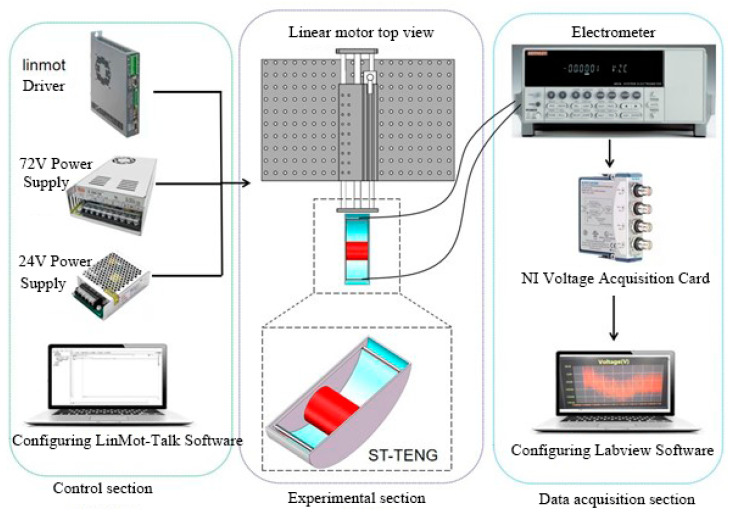
Experimental system diagram.

**Figure 16 micromachines-14-00748-f016:**
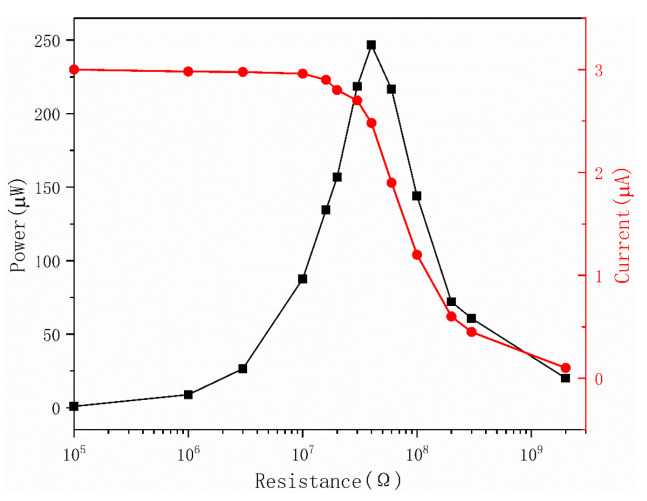
Effect of external load on the output current and instantaneous power of the CS-TENG.

**Figure 17 micromachines-14-00748-f017:**
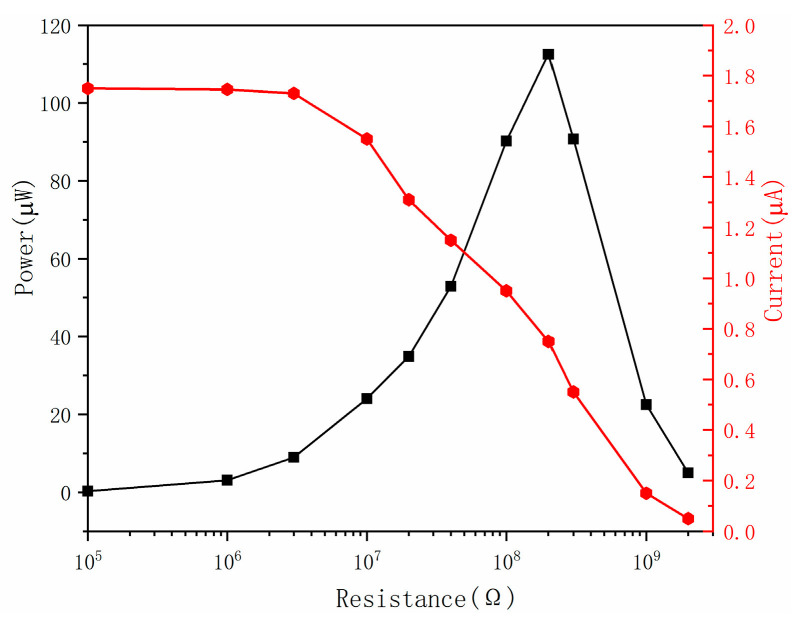
Effect of external load on the F-TENG output current and instantaneous power.

**Figure 18 micromachines-14-00748-f018:**
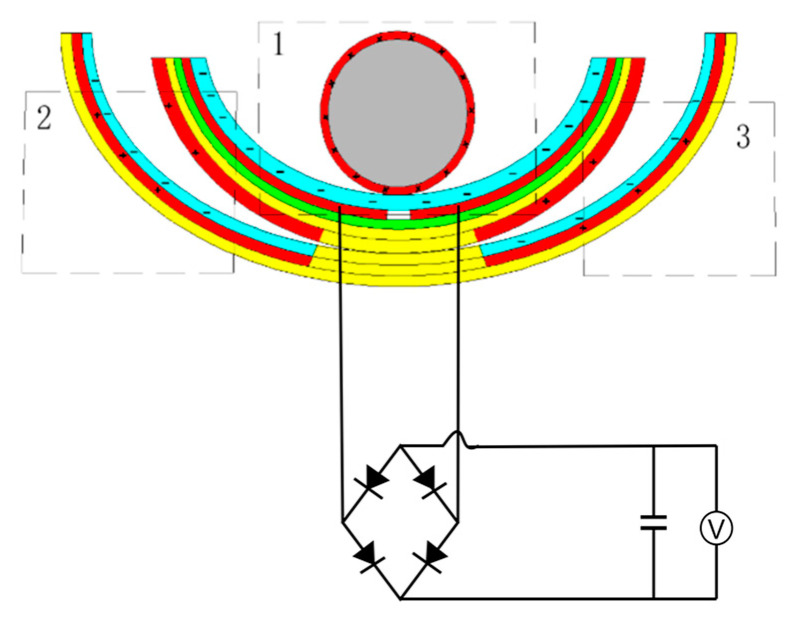
F-TENG connection rectifier charging capacitor wiring diagram. 1 is nylon roller, 2 is built-in boat-shaped device, 3 is external boat-shaped device.

**Figure 19 micromachines-14-00748-f019:**
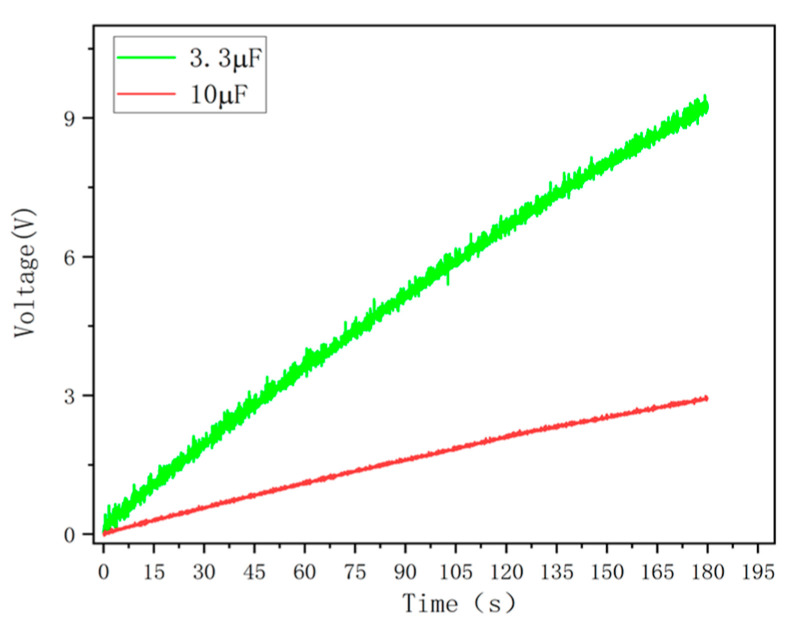
F-TENG charging voltage for capacitors with different capacitance.

**Figure 20 micromachines-14-00748-f020:**
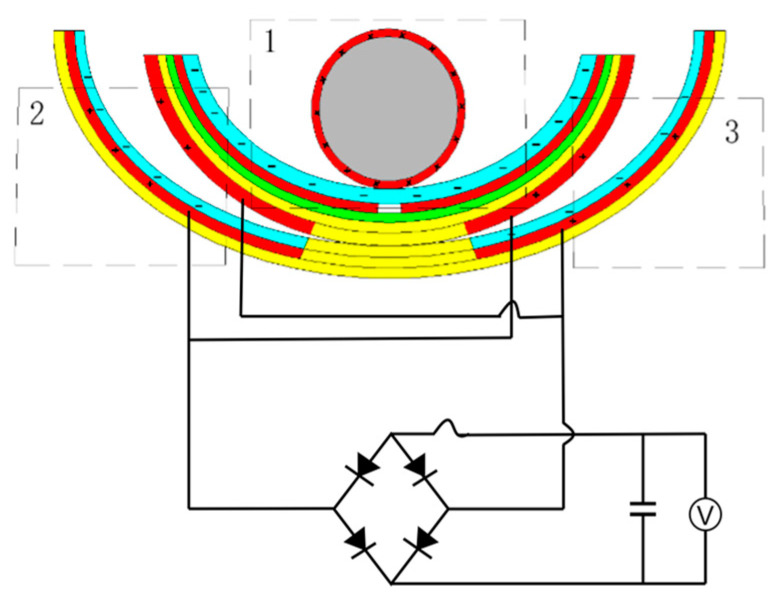
CS-TENG wiring diagram for charging capacitors with a rectifier. 1 is nylon roller, 2 is built-in boat-shaped device, 3 is external boat-shaped device.

**Figure 21 micromachines-14-00748-f021:**
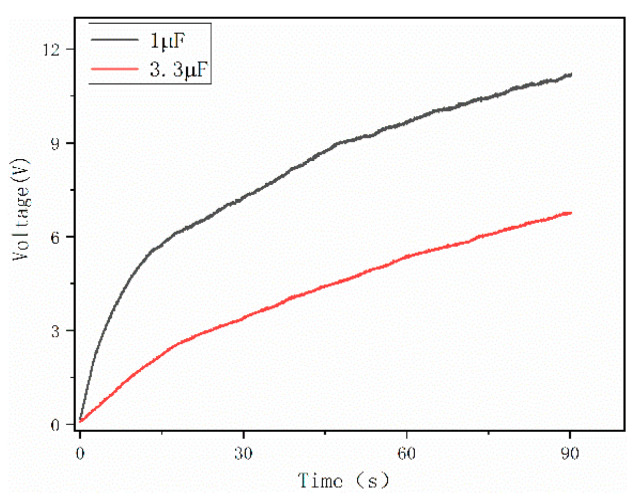
Charging voltage of the CS-TENG to capacitors with different capacitances.

**Figure 22 micromachines-14-00748-f022:**
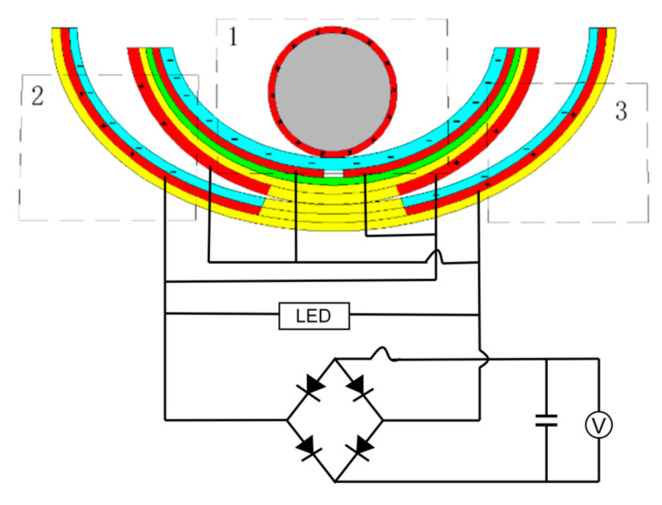
ST-TENG wiring diagram for charging capacitors with a rectifier. 1 is nylon roller, 2 is built-in boat-shaped device, 3 is external boat-shaped device.

**Figure 23 micromachines-14-00748-f023:**
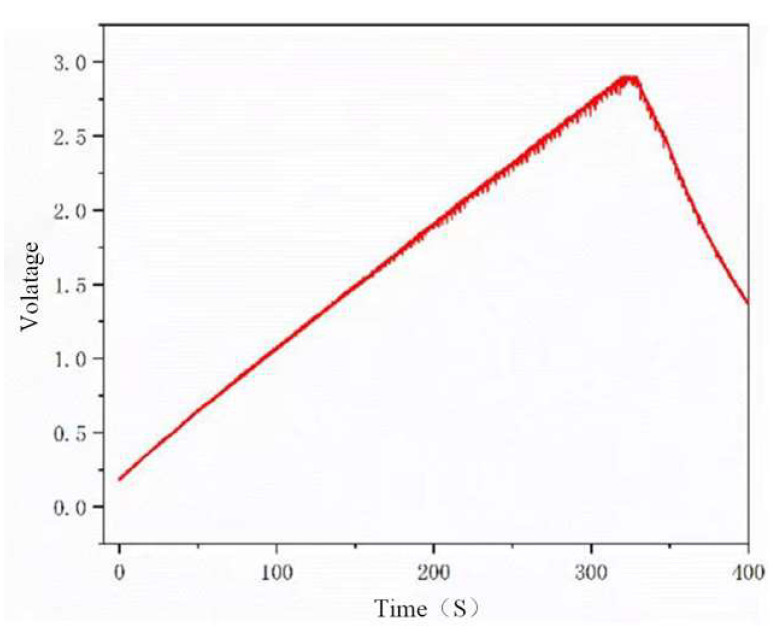
Charging voltage of the ST-TENG to capacitors with different capacitances.

**Figure 24 micromachines-14-00748-f024:**
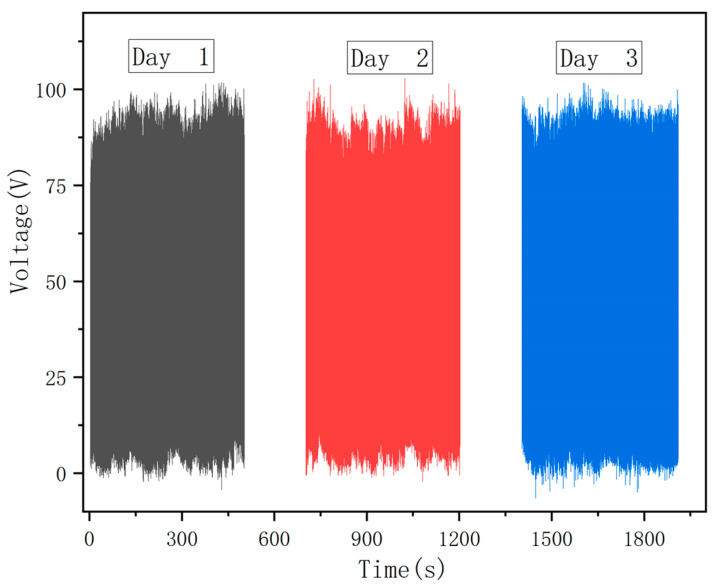
ST-TENG stability experimental test.

**Table 1 micromachines-14-00748-t001:** List of experimental materials.

Name	Specification
Nylon rod	Diameter 50 mm × 70 mm
Polytetrafluoroethylene sheet	0.3 mm thick × 100 mm wide
Polytetrafluoroethylene film	0.08 mm thick × 300 mm wide
FEP	0.08 mm thick × 60 mm wide
Aluminum foil	0.1 mm thick × 50 mm wide
PLA	Diameter 1.75 mm
Foam double-sided adhesive	1 mm thick × 5 cm wide

## Data Availability

Not applicable.

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
