# Peer review of "Triboelectric Nanogenerators for Efficient Low-Frequency Ocean Wave Energy Harvesting with Swinging Boat Configuration"

_micromachines, 2023, doi:10.3390/mi14040748_

Round 1

Author Response

  1. Line 38, the words should be “Maxwell's displacement current”,

Reply: We are very sorry for our incorrect writing. Line 38, the statements of “Max-displacement well's current” were corrected as “Maxwell's displacement current”

  1. There are three parts 1,2,3 displayed in Figure 2b but not mentioned in main text. May authors give more detailed explanation.

Reply: We are very sorry for our negligence of not going into detail in the 123 section of Figure 2(b).We have re-added this section based on the reviewers' comments.

  1. The (c) of Figure 3 covers up the cutlines.

Reply: We have made correction according to the Reviewer's comments.

  1. May the authors add the Unit of the TENG simulation potential distribution of Figure 9 and Figure 12.There articles did outstanding exploration in the area of TENG, which would provide reference for deeding understanding of this article.

Reply: We have re-written this part according to the Reviewer's suggestion. To this end, we add Figure 19 and Figure 14. In response to the comments made by the experts.In response to experts' comments, we have already cited these articles in our paper.

Reviewer 2 Report

This work is trying to study a swinging boat-type triboelectric nanogenerator for efficient low-frequency wave energy collection. The idea is not new. It has been discussed in the research community for quite some years.

First of all, the presentation of the manuscript needs to be improved.

1. What does "efficient low ocean wave enger harvesting" mean? low frequency wave energy?

2. The language needs to be improved. There are some phrases and senstences that don't read correct. Some of them are listed below. It'll be great if the authors can polish the manuscript futher

* page 2, line 38, Max-displacment well's current theory

* page 2, line 41, "cheap cost" should be "low cost"

* page 2, line 42, "a broad variety of materials", why is this a benefit of TENG?

* page 3, line 103

* page 3, line 105

* page 4, line 131, 132

* page 6, line 168

3. The introduction part is not clear. The anthors quoted a lot of work, but it's not clear why the current work is important.

4. There are 26 figures in the manuscript. It's more like a lab report, instead a scientific journal to be published. It'll be great if the authors can make the figures more concise and well-organized. It's also easier for readers to follow the logic. 

Other than the presentation, I also have the following questions.

1. Is the design scalable? Assume the design to be used in the ocean, is there any certain limit to scale the device?

2. Is the any simulation data from the COMSOL simulation to be compared with the experiement? Like to verify the working principle 

3. The generated power is instateneous. How do the authors justify the pratical adoption?

Author Response

Thank you for the reviewers' comments on this article, we will continue to improve, and through this improvement will also lay a good foundation for the future.

  1. What does "efficient low ocean wave energy harvesting" mean? low frequency wave energy?

Reply: Considering the Reviewer's suggestion, we have added line 47  to line 53.

  1. The language needs to be improved. There are some phrases and senstences that don't read correct. Some of them are listed below. It'll be great if the authors can polish the manuscript further.

* page 2, line 38, Max-displacment well's current theory

* page 2, line 41, "cheap cost" should be "low cost"

* page 2, line 42, "a broad variety of materials", why is this a benefit of TENG?

* page 3, line 103

* page 3, line 105

* page 4, line 131, 132

* page 6, line 1683.

Reply: We have re-written this part according to the Reviewer's suggestion. .

  1. The introduction part is not clear. The authors quoted a lot of work, but it's not clear why the current work is important.

Reply: We have made correction according to the Reviewer's comments. As Reviewer suggested that we have added line 55 to line 65.

  1. There are 26 figures in the manuscript. It's more like a lab report, instead a scientific journal to be published. It'll be great if the authors can make the figures more concise and well-organized. It's also easier for readers to follow the logic. 

Reply: We have made correction according to the Reviewer's comments.

5 Is the design scalable? Assume the design to be used in the ocean, is there any certain limit to scale the device?

Reply: Dear experts, hello, thank you for giving us valuable advice in the midst of your busy schedule. Your question is of great significance to this design and can lay the foundation for the subsequent work. The wind energy can be collected by keeping the structure of this design, or by making corresponding modifications, such as using the built-in boat-shaped structure of this design as a circular shell, making the nylon roller of this design into an elliptical structure as the central axis, and using the wind as an external force to generate electricity, and also using ocean wave energy to generate electricity. If the designed product is used in the ocean, the limitation is still relatively small, mainly because of the influence of the ocean, because the wave energy of the ocean is not stable, and the cost of placing the designed product in the ocean depths is higher than that of placing it on the coastline.

6 Is the any simulation data from the COMSOL simulation to be compared with the experiement? Like to verify the working principle .

Reply:COMSOL simulation data should be compared with experiments, as the saying goes, practice is the only standard to test the truth. The combination of simulation data and experiments can make the design more reliable, and each verification of the working principle will make us look forward to it.

7 The generated power is instateneous. How do the authors justify the pratical adoption?

Reply:Through many experiments this designed product power generation can easily drive LED lights and watches, which can already prove the feasibility of its work. In the ocean, the situation is complicated and the power generation is unstable, but the waves are not likely to disappear, and the large-scale use in the ocean ensures that the power generation group can generate enough power when the sea is calm, and the excess power sent out when the sea is not calm can be stored through the battery. Forming a battery to replenish energy when the power is insufficient, and charging energy or converting electricity into other energy such as hydrogen when the energy is excess.

We tried our best to improve the manuscript and made some changes in the manuscript.  These changes will not influence the content and framework of the paper. And here we did not list the changes but marked in red in revised paper.

We appreciate for Editors/Reviewers’ warm work earnestly, and hope that the correction will meet with approval. Once again, thank you very much for your comments and suggestions.

Round 2

Reviewer 2 Report

Thanks for the authors' effort to improve the manuscript. However, the presentation still needs to be improved in the following places.

1. Line 47, there are extra words that are not necesasry - "well's current".

Line 70, Wave energy

Line 114, "As Ah-med et al. A duck-shaped ..." doesn't read correct to me

Overall, more polishing and proof-reading are necessary for the manuscript.

2. The phrase "low ocean wave energy" appears in the title. It's very confusing to me. Do the authors mean "low-frequency ocean wave energy"? This seems to be more commonly used in literature based on my searching. 

3. The indtroduction part still doesn't explain what's new in the current work. What are the pros and cons of the current work compared with the designs in literature. 

4. I don't quite understand the authors' reponsonse to my comment about the comparison between simulation and experiment. Is there any comparison analysis the authors can do?

Author Response

Response: Dear reviewers, your comments are very valuable to us and promote our growth. Based on your comments, we have made serious corrections and hope to receive your approval, and we thank you once again.

  1. Line 47, there are extra words that are not necesasry - "well's current".

Line 70, Wave energy.

Line 114, "As Ah-med et al. A duck-shaped ..." doesn't read correct to me.

Overall, more polishing and proof-reading are necessary for the manuscript.

Reply: We are very sorry for our incorrect writing. We have made correction according to the Reviewer's comments .

  1. The phrase "low ocean wave energy" appears in the title. It's very confusing to me. Do the authors mean "low-frequency ocean wave energy"? This seems to be more commonly used in literature based on my searching. 

Reply: It is really true as Reviewer suggested that is low-frequency ocean wave energy     .

  1. The introduction part still doesn't explain what's new in the current work. What are the pros and cons of the current work compared with the designs in literature. 

Reply: We have re-written this part according to the Reviewer's suggestion. The additions have been marked in blue.

  1. I don't quite understand the authors' reponsonse to my comment about the comparison between simulation and experiment. Is there any comparison analysis the authors can do?

Reply: The simulation analysis shows that the PTFE thickness is 1 mm, Poisson's ratio is set to 0.3, the surface metal of the independent roller and the properties of both electrodes are set to aluminum, the width of both electrodes is set to 1 mm, the radius of the independent roller is set to 3.5 mm, the charge density of the PTFE surface is set to -0.01 μC/m2, and the total charge density of the aluminum surface is set to 0.01 μC/m2. The freestanding TENG can generate a maximum potential difference of 155 V under the condition that the PTFE surface charge density is set to -0.01 μC/m2 and the aluminum surface total charge density is set to 0.01 μC/m2. The dielectric material is set to PTFE with a thickness of 1 mm, both electrodes are set to aluminum material with a width of 1 mm, the surrounding material is set to air, and the relative dielectric constant is 1. The same PTFE surface charge density is set to -0.01 μC/m2 and the total charge density on the aluminum surface is set to 0.01 μC/m2. the contact TENG can generate a maximum potential difference of 100 V, confirming the feasibility of this design to generate electrical energy, laying the foundation for the next experiments. The ST-TENG can charge the 33µF capacitor to 3V in 320s and keep the electronic watch in normal operation for 45s. There is a progressive relationship between the specific experiments, and the differences in the data are highly related to the parameters we set. The specific power generated by this design is directly related to the magnitude of the wave energy. The simulation experiment proves that the design can generate electricity in theory, but the parameters of the simulation experiment are artificially set to the ideal size of the parameter electricity, and the experiment finds that the size of the parameter electricity is still related to the frequency of wave energy. However, both the simulation and the specific experiments are sufficient to prove the feasibility of this design for powering marine equipment.

We tried our best to improve the manuscript and made some changes in the manuscript.  These changes will not influence the content and framework of the paper. And here we did not list the changes but marked in blue in revised paper.

We appreciate for Editors/Reviewers’ warm work earnestly, and hope that the correction will meet with approval.

Once again, thank you very much for your comments and suggestions.
